# Residue Cluster Classes: A Unified Protein Representation for Efficient Structural and Functional Classification

**DOI:** 10.3390/e22040472

**Published:** 2020-04-20

**Authors:** Fernando Fontove, Gabriel Del Rio

**Affiliations:** 1C3 Consensus, Miguel Hidalgo, CDMX, Mexico City 11510, Mexico; fernando.fontove@c3consensus.com; 2Department of Biochemistry and Structural Biology, Instituto de Fisiología Celular, UNAM, Mexico City 04510, Mexico

**Keywords:** residue cluster class, structural classification, functional classification

## Abstract

Proteins are characterized by their structures and functions, and these two fundamental aspects of proteins are assumed to be related. To model such a relationship, a single representation to model both protein structure and function would be convenient, yet so far, the most effective models for protein structure or function classification do not rely on the same protein representation. Here we provide a computationally efficient implementation for large datasets to calculate residue cluster classes (RCCs) from protein three-dimensional structures and show that such representations enable a random forest algorithm to effectively learn the structural and functional classifications of proteins, according to the CATH and Gene Ontology criteria, respectively. RCCs are derived from residue contact maps built from different distance criteria, and we show that 7 or 8 Å with or without amino acid side-chain atoms rendered the best classification models. The potential use of a unified representation of proteins is discussed and possible future areas for improvement and exploration are presented.

## 1. Introduction

Proteins are molecules found in living organisms and participate in many diverse cellular and molecular functions. It is generally recognized that the three-dimensional (3D) structures of proteins are related to their functions, yet the relationship remains to be elucidated, since many attempts to predict the correct functions of proteins based on their 3D structures still result in false predictions [1].

Different approaches have been described to account for such a relationship, including those based on physical and chemical forces [2,3], protein sequence and phylogenies [4,5,6] and others. Machine-learning (ML) based models are currently the best models for predicting 3D protein structures [7] and protein functions [8]. ML models require object representations in the form of a set of features; these features are numeric values; hence, objects are represented by vectors. Yet, while protein structure and function are indeed related (both observations are derived from the same object, the protein, and hence are related), it is questionable whether at the model level these are actually related. For instance, the features used so far to predict the 3D structure are different from those used to predict protein function [9,10]; hence, the reliability of ML methods when predicting a protein’s structure or function may be simply the consequence of having better representations for each aspect of the protein (i.e., 3D structure or function). We believe having a unique protein representation with which to efficiently predict 3D protein structure and function would provide a mathematical framework to explore the relationship between these two fundamental aspects of proteins, 3D structure and function. 

We have previously described a representation of 3D structure that is learnable (that is, it displays a pattern that any ML heuristic model should be able to detect); such a representation allowed us to identify structural neighbors and classify the protein’s 3D structure with the best performance and reliability reported so far [11]. The representation is based on counting the 26 different maximal clique classes that are derived from the 3D structure and protein sequence given a contact distance threshold of 5 Å, including atoms of the side chains; we referred to these maximal cliques as residue cluster classes or RCCs (see Figure 1 and Materials and Methods). In that previous work, we tested two ML algorithms (random forest and support vector machine) that were adequate for the structural classification of the data. In the present work, we developed a computationally efficient implementation for computing RCCs on large datasets of 3D protein structures. This allowed us to further explore the protein structure classification and distribute this implementation freely (see Appendix A); this implementation incorporates some variations in the contact definition (different distances and the exclusion of the side-chain atoms). Here we also explored in a systematic way for dozens of ML models to identify the optimal model and corresponding hyper-parameters for such tasks, and corroborated that random forest rendered the best algorithm. Furthermore, we showed that ML models built from RCCs were able to efficiently learn protein functions, yet there is plenty of room for improvement. Hence, our results provide the first unified, high-dimensional description of proteins useful for learning both 3D structure classification and protein function using ML methods.

## 2. Materials and Methods

### 2.1. RCC Calculation Implementation

The pseudo code to obtain the RCC from PDB entries is depicted in Algorithm 1:
 **Algorithm 1: RCC calculation** Input: Cartesian coordinates of protein atoms Output: RCC coordinates 1: Generate graph with the contacts of all residues. Two residues are in contact if they are within a distance threshold. 2: Calculate maximal cliques using Tomita algorithm (see below) 3: Calculate RCC from maximal cliques.

#### 2.1.1. Contact Map Calculation

We accelerated the calculation of all contacts given a distance threshold *d* with the use of several geometric restrictions:

(i) The hashing approach using a 3D grid over the space. We generated a grid over the 3D space of width *d* that corresponded with the contact threshold distance; thus, given a residue we can identify the cube that contains it by using a hashing function; every cube in the grid has a list of all the residues that are inside of it. Given a residue *r* and the cube in the grid where it is found, *G_r_*, we know that any other residue inside *G_r_* must be in contact with *r*. Additionally, if a residue *s* is in contact with *r*, it must be that the cubes *G_r_* and *G_s_* are neighbors in the grid (if they were not, then *r* and *s* must be at a distance greater than *d*). This way we reduced the search space dramatically (see Figure 2), which turns the complexity of the algorithm to 2**n* (*n* is the number of amino acids in a protein) in the linear portions of the protein and 16**n* on average (given that at 5 Å a residue is in contact with four other residues on average).

(ii) Rules for quick evaluation (see Figure 3). For a residue *r* we enveloped it in a sphere of radius *r_d_* and center *r_c_*. For two residues *r* and *s*, let *D* be the distance between *r*_c_ and *s*_c_,. They must be in contact if *D* < *d*,(1) and cannot be in contact if minimum {*D* − *r_d_*, *D* − *s_d_*} > *d*.(2)

Finally, they must be in contact if a sphere of radius *d* and center *r_c_* overlaps half the volume of the sphere enveloping *s* or *vice versa* (though this last rule was turned off in the final implementation due to the overhead of the calculation being too similar to the time saved by it). Any pair of residues that do not fulfill one of these rules needs the explicit computation of the distance between its pair of atoms to be performed (with an early stop if a contact is found).

In the scenario in which the side chains are ignored, the spheres are calculated with the remaining atoms of the residue. The geometric restrictions work in the same way as if the spheres were calculated with all the atoms in the residue, and similarly, when a pair of residues does not meet the criteria for quick evaluation, the explicit computation of the distances between their pair of atoms is performed exclusively with the atoms in the backbone. The pseudo-code for this calculation is detailed in Algorithm 2:
**Algorithm 2: Contact map calculation**  Input: Cartesian coordinates of protein atoms  Output: Contact graph  For each residue *r* in the protein:   Calculate its cube in the grid *G_r_* using the hash function.   Get the list of residues *L* inside the cubes neighboring *G_r_*.   For each residue *s* in *L*:    Calculate *D* distance between *r_c_* and *s_c_*.    Quick inclusion if *D* < *d* and continue.    Quick exclusion if minimum {*D*–*r_d_*, *D*–*s_d_*} > *d* and continue.    Perform contact check for each pair of atoms from *r* and *s*. 

#### 2.1.2. Maximal Cliques Calculation

Calculating the maximal cliques from a graph is a classical problem in computer science presented by Tomita [14], and the fastest implementation to our knowledge by Eppstein and Strash [15] was used in our method. To use said implementation, we converted the residue identifier to an index starting at 0; hence, the first residue in a 3D protein structure is labeled 0, the second 1 and so on.

#### 2.1.3. RCC Calculation

We only considered maximal cliques containing at least three residues and at most six residues, which are grouped in 26 cluster classes. This grouping for a given protein produces different frequencies for each of these 26 clusters, wherein the first 3 frequencies correspond with maximal cliques with 3 residues ([1,1,1], [1,2] and [3]); the next 5 frequencies are maximal cliques with 4 residues ([1,1,1,1], [1,1,2], [2,2], [1,3] and [4]); the next 7 frequencies are maximal cliques containing 5 residues ([1,1,1,1,1], [1,1,1,2], [1,1,3], [1,4], [1,2,2], [2,3] and [5]); and the last 11 frequencies include maximal cliques with 6 residues ([1,1,1,1,1,1], [1,1,1,1,2], [1,1,1,3], [1,1,2,2], [1,1,4], [1,2,3], [1,5], [2,2,2], [2,4], [3,3] and [6]). The number of residues that are adjacent in the protein sequence define the class of a residue cluster. For instance, an RCC with 3 residues in which all residues are not adjacent in the protein sequence is referred to as [1,1,1]; an RCC with 4 residues in which 2 residues are adjacent in the protein sequence, and other 2 residues are also adjacent in the protein sequence (e.g., residues 45 and 46, and residues 101 and 102) is represented as [2,2] (see Figure 1). The regions separating these non-adjacent residues in the protein sequence could include different secondary structure elements (e.g., loops, helices). The maximum size of cliques is limited to 6 due to larger numbers being extremely rare at 5 Å and was kept as is in all the runs in order to keep the results comparable and avoid potential over-fitting.

### 2.2. RCC Database

For the 3D structural classification, we obtained the RCC for every protein domain reported in CATH (version 4.2.0) using 7 different distance cut-off values (5, 6, 7, 8, 9, 10 and 15 Å) and including or not the atoms of the amino acid side chain; hence, for each protein domain, 16 RCC representations were obtained. Our RCC dataset includes 354,079 different proteins (we noted that CATH 4.2 included 434,857 different domains, yet for several of these there was no PDB associated, so not all CATH domains are included in our dataset). A total of 2,831,584 different RCCs corresponding to those representations with amino acid side-chains or without side (see Appendix A).

For the functional classification, we used the Gene Ontology (GO) database (version 2.0, generated on October 2017) and calculated the RCCs for every protein in the PDB database [16]. Similarly to the 3D structure, we used the 7 cut-off values and inclusion or exclusion of the side chains. By cross-referencing the 869,535 functions reported in GO (C: 176,437; F: 422,681; P: 270,416) with the 354,079 protein domains in the PDB, we obtained a database of 4,991,252 annotated proteins (C: 1,192,742; F: 2,2,85,509; P: 1,513,001) with their corresponding RCCs and known protein functions. The total number of annotated proteins is higher than the PDB protein domain because several proteins have multiple annotated functions, and a single protein sequence may have multiple chains in a single PDB file. To calculate the RCCs for all these PDB entries, we used our code and execute it on a 64-bit Intel-Xeon linux-based server with 24 cores and 256 GB of RAM.

### 2.3. Model Training and Testing

For identifying a model to classify protein structures, the full set of RCCs was labeled according to the annotated CATH classification in each of its 2 levels: 4 Classes (all alpha; all beta; a mixture of alpha and beta; or little secondary structure) and 41 Architectures. This full set was used to train models explored using the AutoWeka plugin [17], which performs an optimization over the ML models included in Weka (J48, DecisionTable, GaussianProcess, M5P, Kstar, LMT, PART, SMO, BayesNet, NaiveBayes, JRip, SimpleLogistic, LinearRegression, VotedPerceptron, SGD, Logistic, OneR, MultilayerPerceptron, REPTree, IBk, M5Rules, RandomForest, RandomTree and SMOreg; and the meta classifiers, which combine the previous models in different ways: Vote, Stacking, Bagging, RandomSubSpace, AttributeSelectedClassifier and RandomCommittee), their hyper-parameters and their select attributes (BestFirst, GreedyStepwise and CfsSubsetEval); AutoWeka was executed first for 20 minutes and identified as the best models those built using RCC at 7 or 8 Å with our without side chains. These same models were further analyzed under AutoWeka for 24 hours. In all these cases, random forest with hyper-parameters unlimited tree depth, no attribute selection and 100 iterations rendered the best results. The rest of the training datasets were run using this algorithm and hyper-parameters on a 64-bit Intel-Xeon linux-based server with 24 cores and 128 GB of RAM. Finally, 10-fold cross validation using the WEKA package was also performed to all these training sets [18].

The statistical parameters (accuracy, precision, correctly classified instances) reported by Weka and AutoWeka were used to evaluate the learning performance of the classifiers.

## 3. Results

The code implementing an efficient computation of residue cluster classes (RCCs) is freely (see Appendix A); the computational efficiency of this implementation is reported in Table 1.

This computational efficiency allowed us to compute RCC values for all protein structures in the PDB database (354,079 protein domains [19]) in less than 1 hour (see Figure 4) on a 64-bit Intel-Xeon linux-based server with 24 cores and 128 GB of RAM. Considering that the PDB database includes about 10,000 new entries every year, which tend to include 200 residues each, this implementation would be able to compute these new entries in 11 minutes on average.

### 3.1. Protein Structural Classification

We built different RCC representations of protein structures to identify the best distance criterion to classify 3D protein structures. These included distances at 5, 6, 7, 8, 9, 10 and 15 Å; we originally used only a distance of 5 Å [11]; hence, this exercise allowed us to compare the efficiency of RCCs previously reported. In addition, we also included a variant in the construction of RCCs: the inclusion or exclusion of the amino acid side-chain atoms. This variation reproduces the preponderant role of the backbone in visual protein structure classification and consequently would allow for testing whether this representation is sufficient to learn 3D protein structure classification. Finally, we searched for the best model using an automatic approach based on the optimization algorithm implemented in AutoWeka, and to test the reliability of our model’s classification, we conducted cross-validations; the labels of structural classes of proteins were derived from the CATH database (see Methods and Methods).

We observed that the best models were obtained using RCC representations with a contact distance threshold of 7 or 8 Å without side chains (see Figure 5). Table 2 summarizes the best model performance compared with previous results by Corral and collaborators.

### 3.2. Protein Functional Classification

We used the same protein structure representations described for structural classification for protein function classification; to include or not the side-chain atoms here represent an exploration of the relevance of side-chain contacts for protein function. For the functional annotation, we used the Gene Ontology (GO) annotations, consisting of three main classes: molecular function (F), biological process (P) and cellular localization (C). As described before, AutoWeka and cross-validation were used to identify the best models to learn GO functional annotation from RCC representations.

The best models were again observed with 7 or 8 Å of distance between the atoms of residues with or without side-chain atoms (see Figure 6). Table 3 summarizes the best model performance compared with the statistics reported for the second version of the Critical Assessment of protein Function Annotation contest (CAFA2) for the best models [20]. Fmax is the maximum harmonic representation of the precision and recall achieved by a set of models; Fmax = 1 is for a perfect predictor. It is important to note that the CAFA experiment attempts to unify the prediction of protein function, but does not use 3D protein structure; hence, 15% of all sequences included in CAFA2 were included in our datasets (data not shown). The best models in CAFA2 used sequence alignments and ML models that incorporated diverse proteins features, while our models only used RCCs—features derived exclusively from the protein structure. The purpose of this comparison is not to show better performance than CAFA2 models, but to note the level of reliability of our predictions in comparison with the function predictors known to be the best; that is, the best predictors in CAFA2 were close to Fmax = 0.5, so to our models. Hence, these results show that RCC achieved reliable classifications in both protein structure and protein function.

An important aspect in functional annotation is the biased compositions of different classes; such a bias may cause predictors to classify proteins according to the most abundant functional classification. To rule out the possibility that such a bias may have affected the reliability of our predictions, we conduced a set of calculations using the ZeroR classifier in Weka; this classifier only predicts the most frequent class; hence, any machine-learning algorithm must achieve a performance better than this value to be reliable. In Figure 7 (A: cellular localization; B: molecular function and C: biological process) we show the percentages of correctly classified instances by the ZeroR classifier (square symbols) in comparison with those achieved by the best models (circle symbols) using RCCs built with different distance criteria. All the best models were above the baseline ZeroR predictions.

## 4. Discussion

Including the efficient implementation of Tomita’s algorithm into the RCC calculation rendered a constant time performance independent of the protein lengths analyzed; we also noted that the time required to compute the contacting distances and get the RCCs was less than reading the input file. In terms of computing time, assuming these calculations would be done on a single central processing unit or CPU, the global time required to compute a given RCC is divided in the time required to load the protein’s 3D structure or PDB file (elapsed time) and the time to obtain the RCC (CPU time), which includes constructing the contact map of residues and identifying the maximal cliques. Our results indicate that most of the time is taken by reading the PDB file; this may be improved by implementing a non-serial reading function [21]. This in turn would benefit from the improvement of current CPU technologies [22]. Additionally, considering that computing RCCs from a large database is an embarrassingly parallel problem and our current code has been implemented to deal only with concurrent computing (local computing, as opposed to distributed computing), we anticipate that there is still room for improvement in terms of using a distributed computing scheme, such as the ACTOR formalism [23].

The latest Critical Assessment for Structure Prediction (CASP) competition for the first time showed that ML (AlphaFold) improved on previous methods using protein sequence features [24], yet the features used by AlphaFold are not the same used by the best models of protein function reported in CAFA contest [8]. Thus, RCC is the first representation of proteins that allows for the efficient modeling of both fundamental aspects of proteins. 

Our screening to build RCCs reveals that the backbone contains enough information to represent both 3D structure and function. This does not imply that side chains are not relevant for 3D protein structure or function; after all the backbone conformation depends on the side chains. The relevance of building RCCs without side chains is that on the one hand, such a representation does not need high-resolution structures to build a useful model; this would be relevant to further exploration: what is the range of protein structure resolution that renders a reliable model for protein structural and functional classification? On the other hand, having a model that concerns only on the protein backbone may facilitate the development of methods to predict 3D protein structure based on RCCs.

Having the same representation of proteins to model 3D structure and function would allow one to analyze the possible co-localization of structural and functional classes in the 26-dimensional space of RCCs, for instance; this would eventually lead to a better understanding of the structure–function relationship of proteins. RCCs would allow exploring for regions in this 26-dimensional space where no examples of 3D protein structure or function are known, and potentially, designing new proteins.

In summary, in this work we distribute a computationally efficient implementation with which to compute RCCs from a 3D protein structure—a representation of proteins that allows for effective modeling of both 3D protein structure and functional classification.

## Figures and Tables

**Figure 1 entropy-22-00472-f001:**
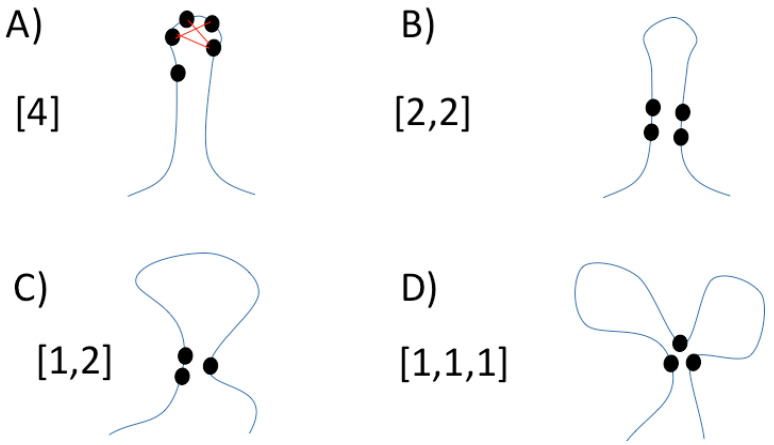
**Residue Cluster Classes**. Folded proteins (represented by blue lines) highlighting residues (black circles) in contact forming different classes of maximal cliques; a clique is a set of nodes (in the case of proteins, nodes are amino acid residues) wherein all nodes are in contact with each other; a maximal clique is that clique that is not part of another clique. (**A**) Maximal clique of class [4], where four sequence adjacent residues form a clique; these four residues are indicated on the top of the image connected by red lines (edges) and cannot be extended to form a clique of size five with another residue. (**B**) Maximal clique of class [2,2], where two sequence adjacent residues form a clique with other two adjacent residues; the four residues are not adjacent. (**C**) Maximal clique of class [1,2], where two sequence adjacent residues form a clique with a separated third residue. (**D**) Maximal clique of class [1,1,1] where three sequence non-adjacent residues form a clique. The regions separating these non-adjacent residues in the protein sequence could include different secondary structure elements (e.g., loops, helices).

**Figure 2 entropy-22-00472-f002:**
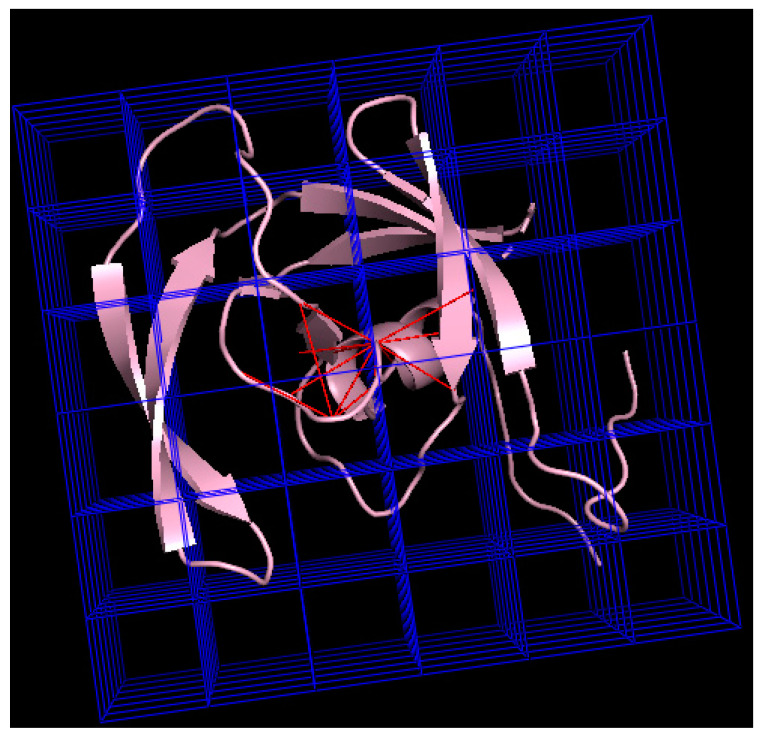
**Hash-like approach to build a contact map**. A grid is placed on top of the protein; its width is the contact distance. At the center, the residue of interest and the red lines connecting the residue of interest with the close by residues. Any residue in contact with the residue of interest must be in a neighboring cube; if it is inside the same cube, it must be in contact. The image was generated using CMView (version 1.1.1) [12], and the PyMol script DrawGridBox [13].

**Figure 3 entropy-22-00472-f003:**
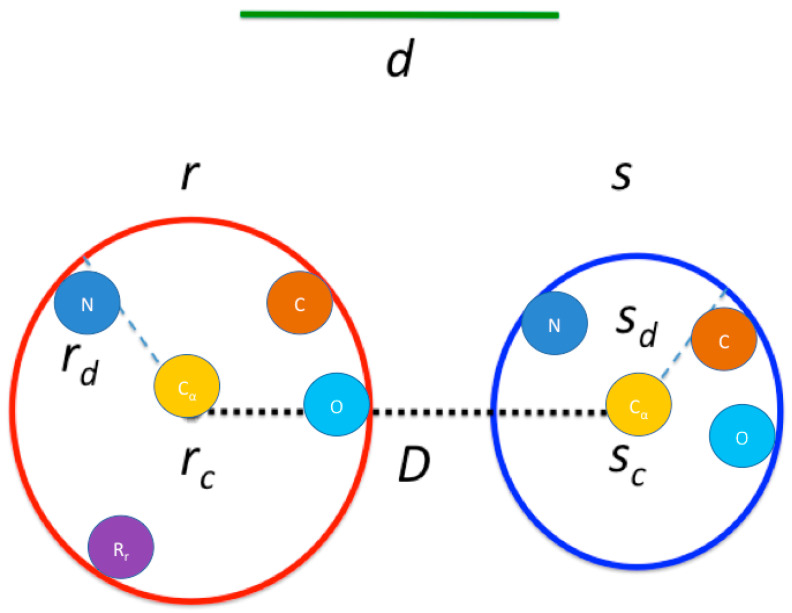
**Contact map algorithm**. *r* and *s* are two residues potentially in contact; each is enveloped in a sphere with center *r*_c_ or *s*_c_, and radius *r*_d_ or *s*_d_, respectively. The contact distance is *d* and the distance between *r*_c_ and *s*_c_ is *D*. For each sphere, the center is calculated as the geometric center for all the atoms in the residue, including the side-chain atoms (the center does not necessarily overlap with any atom), and the radius is the distance between the center and the farthest atom, which may vary depending on the length and structure of the side chain. In the case in which the side chain is ignored, the spheres may still be not identical due to the elasticity of the bonds between the atoms (i.e., the distance between the nitrogen and carbon alpha atoms is not exactly constant, just like the internal angles of the backbone).

**Figure 4 entropy-22-00472-f004:**
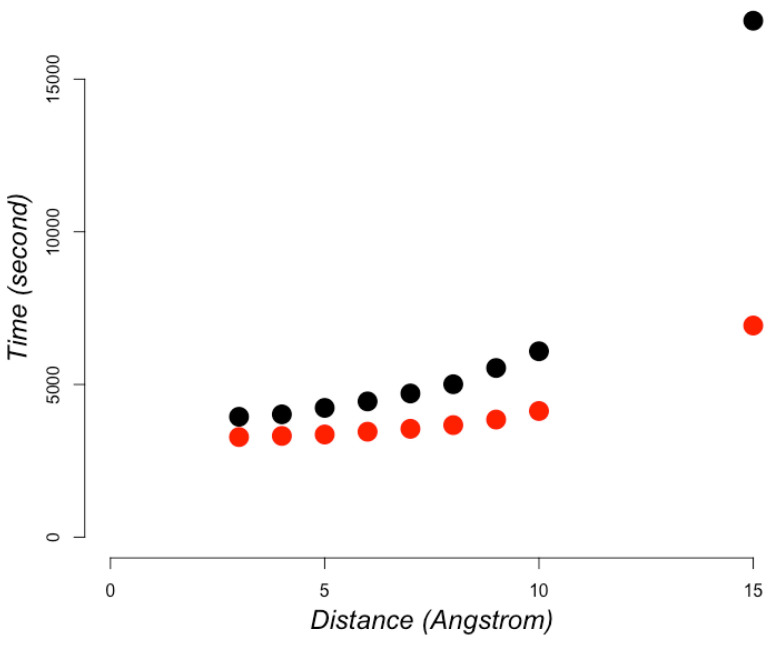
Time to compute RCC in PDB dataset. Computing 354,079 protein domains as annotated in CATH database including (black) or excluding (red) the atoms of the amino acid residue side-chains.

**Figure 5 entropy-22-00472-f005:**
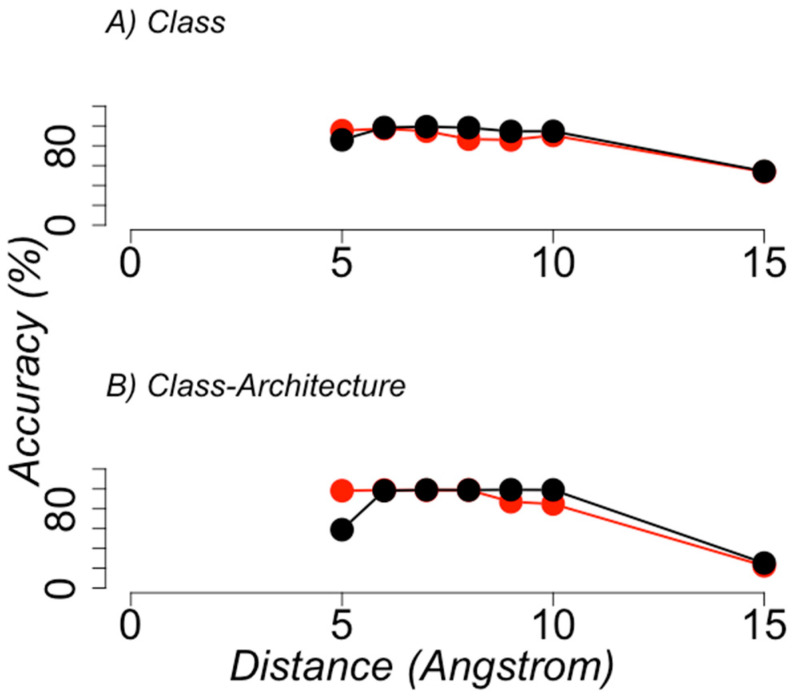
**Best classifier performance on the structural classification of proteins**. The average accuracy (y axis) achieved by the best classifiers after a 10-fold cross-validation test (see Methods and Methods) is shown for the different distance cutoff values used to build RCC (x axis), in the task of annotating the CATH structural classification of proteins. RCCs built using side-chain atoms are shown in red circles; RCCs built without side-chain atoms are shown in black circles. (**A**) Shows the performance when learning the class annotation from CATH classification, and (**B**) the class-architecture annotation from CATH classification.

**Figure 6 entropy-22-00472-f006:**
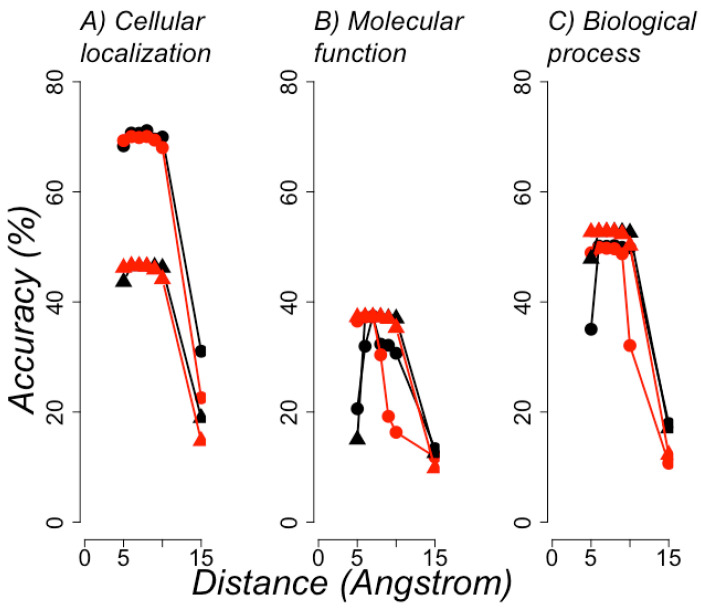
**Best functional classification of proteins**. The average accuracy (y axis) achieved by the best classifiers after a 10-fold cross-validation test (see Methods) is shown for the different distance cutoff values used to build RCCs (x axis), in the task of annotating the GO functional classifications of proteins. Results obtained with RCC built using side chains are in red and those without side chains in black; circle symbols indicate that functional annotation was done for all proteins; triangle symbols are for proteins with single domains. (**A**) Presents the results for predictions of cellular localization; (**B**) molecular function; (**C**) biological process, as annotated in GO.

**Figure 7 entropy-22-00472-f007:**
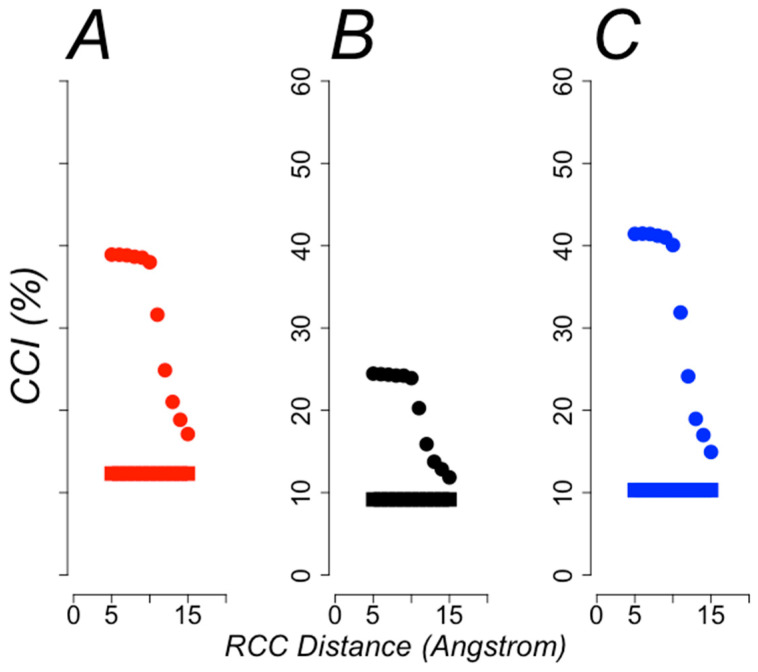
**Baseline performance for the best models**. The percentage of correctly classified instances (CCI (%)) is plotted for every model built from RCCs without side chains at different distances (x axis). The baseline performance achieved by ZeroR classifier (square symbols) is compared with that observed for the best models (circle symbols). (**A**) The comparison for cellular localization (red symbols), (**B**) for molecular function (black symbols) and (**C**) for biological process (blue symbols).

**Table 1 entropy-22-00472-t001:** Computer efficiency of our RCC implementation.

PDB ID*	Length**	Reading*** (ms)	Graph Calculation (ms)	Maximal Cliques (ms)	Total Time (ms)
1ORN (A)	214	39	19	6	64
2HOX (A)	425	110	37	6	153
3GVK (A)	644	202	47	6	255
1F8N (A)	818	329	67	6	402

* The letter in parentheses corresponds with the chain used for the listed PDB entries. ** Number of amino acid residues in the protein analyzed. *** Time taken in milliseconds (ms) reading the PDB file.

**Table 2 entropy-22-00472-t002:** Parameters for best 3D protein classification.

CATH Level	Mean Cross-Validation Accuracy * (Corral et al)	Mean Cross-Validation Accuracy ** (Current)
C	0.96	0.98
A	0.88	0.89

* This accuracy was reported using a random forest implementation in Sklearn in Python language. **This accuracy was obtained using a random forest implementation in Weka in Java language.

**Table 3 entropy-22-00472-t003:** Parameters for best protein functional classification.

GO Function	CAFA2* Fmax	Fmax**	Fmax***
C	0.46	0.44	0.58
F	0.59	0.24	0.48
P	0.37	0.41	0.54

* This accuracy was obtained from the reported Fmax values of the second version of CAFA [20]. ** This Fmax value corresponds to all proteins with an RCC computed in this study. *** This Fmax value considered only proteins with a single chain in PDB or single domain proteins.

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
