# Peer review of "Residue Cluster Classes: A Unified Protein Representation for Efficient Structural and Functional Classification"

_entropy, 2020, doi:10.3390/e22040472_

Round 1

Reviewer 1 Report

The authors propose the use of Residue Cluster Classes (RCC) in ML-based modeling for structure and function classification of proteins.

Three big concerns/questions.

Firstly, what benefit is gained in calculating RCC over simply the use of the PDB files themselves? A PDB file contain the entirety of the (resolved) 3D structure of a biomolecule, and the residue sequence in a PDB file is the sequence which can be used for the purpose of gene ontology studies. You are taking a rich data set (PDB), from which you calculate a meta metric (RCC), which then is fed into an ML model, which you show can recoup structural and functional details as related to a genomic analysis. ML is at the point where more and more models -- especially deep learning -- perform very well given enough data points. Eventually, throw enough data into this-and-that ML classifier, and it will get good results. In your case, indeed you have a large data set.

Secondly, AutoWEKA is used, but wholly as a black box. No details are provided. What is the set of models from which AutoWEKA selects the best model? Regarding hyperparameter optimization, that is usually a very costly and time intensive approach. You tout speed of calculating RCC, but then do not mention the hyperparameter optimization step, which if it requires hours or days, does calculating RCC in seconds really matter? 

Thirdly, you provide results in the form of accuracy values per the use of RCC in being able to recoup structural and functional information about proteins. But how does the use of RCC compare to the use of, say, just course-grained PDB data in various ML-based methods to recoup structure classification and gene ontology details?

You mention that RCC "is able to learn protein function." Don't you mean that an ML model is able to learn?

The pseudocode in 2.1 lacks any reasonable amount of detail. Three lines only? And how do you generate a graph? What are the details?

The run times for calculate RCC are impressive, but you don't mention compute resources. Use of HPC? Quad core? GPUs?

The use of the term "hash-like" is ambiguous. Hashing in the context of computer science refers to hash functions, which for perfect hashing are O(constant). I think there is cross-discipline terminology confusion here that should be cleared up. And what do you mean by "like." Why not call it Grid-like?

For the hash-like approach, what is the cost of identifying if atoms are in the same grid? You don't provide details, and it seems like your hash-like approach is a magical fix for everything.

For your quick rules evaluation approach, how does that improve on n^2? The intro to section 2.1.1 mentions n^2, so please explain further.

Why at most are 6 residues considered? In a sphere of size 15 Angstroms, is that the maximum number of residues that can fit into that space?

Does [1,1,1,1,1,1] refer to a long loop? It would be good to get a better sense of these, much like you did for [4], [2,2], [1,2].

You mention a variant -- use of side chain atoms -- of RCC. But you never mention at the onset that your "original" RCC is coarse grained, which I presume is CA only.

Grammar-wise, often times articles ("the" and "a") are missing. 

I like the figures, but the text in them is too big, suggesting a not-too-careful cut and paste from a different manuscript approach.

Author Response

1) Firstly, what benefit is gained in calculating RCC over simply the use of the PDB files themselves? A PDB file contain the entirety of the (resolved) 3D structure of a biomolecule, and the residue sequence in a PDB file is the sequence which can be used for the purpose of gene ontology studies. You are taking a rich data set (PDB), from which you calculate a meta metric (RCC), which then is fed into an ML model, which you show can recoup structural and functional details as related to a genomic analysis. ML is at the point where more and more models -- especially deep learning -- perform very well given enough data points. Eventually, throw enough data into this-and-that ML classifier, and it will get good results. In your case, indeed you have a large data set.

In order to run the ML methods, features need to be extracted from the raw data (in this case the PDB). In this sense, the RCC is not a meta metric per se, but a set of features derived from the PDB, on which we can then run a ML method. To clarify this, we have included in the Introduction section “ML models require object representations in the form of a set of features; these features are numeric values hence objects are represented by vectors”.

For better performance, ML models require large datasets, and the accumulation of PDB entries allow now the efficient use of ML approaches to model protein structure and function, as we show in this work. However, it is not trivial to identify the correct ML model to classify either the structure or function of proteins. Our work does a systematic exploration of all possible ML models implemented in one of the most complete suites of ML models (Weka) to identify those models and corresponding hyper-parameters that best fit the data. To clarify this, we have included in the Introduction section the following sentence: “…In that previous work, we tested two ML algorithms (random forest and support vector machine) which were adequate to the structural classification of the data. In the present work, we developed a computationally efficient implementation to compute RCC on large datasets of protein 3D structures. This allowed us to further explore the protein structure classification and distribute this implementation freely (see https://github.com/C3-Consensus/RCC); this implementation incorporates some variations in the contact definition (different distances and the exclusion of the side-chain atoms). Here we also explored in a systematic way dozens of ML models to identify the optimal model and corresponding hyper-parameters for such tasks and corroborated that random forest rendered the best algorithm…”.

2) Secondly, AutoWEKA is used, but wholly as a black box. No details are provided. What is the set of models from which AutoWEKA selects the best model? Regarding hyperparameter optimization, that is usually a very costly and time intensive approach. You tout speed of calculating RCC, but then do not mention the hyperparameter optimization step, which if it requires hours or days, does calculating RCC in seconds really matter? 

AutoWeka is a method inside the Weka package that performs optimization over the available models, hyperparameters and attribute selection. As an optimization process, one of the criterions to stop the search procedure is the desired runtime for the optimization. The default is to run it for 20 minutes, we allowed it to run for up to 24 hours and the results were the same: Random forests with default parameters (unlimited tree depth, no randomly chosen attributes, 100 iterations, 100% of the trained set used for bags) and no attribute selection.

The models available to AutoWeka are: J48, DecisionTable, GaussianProcess, M5P, Kstar, LMT, PART, SMO, BayesNet, NaiveBayes, JRip, SimpleLogistic, LinearRegression, VotedPerceptron, SGD, Logistic, OneR, MultilayerPerceptron, REPTree, IBk, M5Rules, RandomForest, RandomTree, SMOreg, and the meta classifiers (these combine the previous models in different ways) Vote, Stacking, Bagging, RandomSubSpace, AttributeSelectedClassifier, RandomCommittee. For attribute selection it has: BestFirst, GreedyStepwise and CfsSubsetEval.

The text of section 2.3 has been changed to explain this: “…using the AutoWeka plugin [17]⁠, which performs an optimization over the ML models included in Weka (J48, DecisionTable, GaussianProcess, M5P, Kstar, LMT, PART, SMO, BayesNet, NaiveBayes, JRip, SimpleLogistic, LinearRegression, VotedPerceptron, SGD, Logistic, OneR, MultilayerPerceptron, REPTree, IBk, M5Rules, RandomForest, RandomTree, SMOreg, and the meta classifiers, that combine the previous models in different ways: Vote, Stacking, Bagging, RandomSubSpace, AttributeSelectedClassifier, RandomCommittee), their hyper-parameters and select attributes (BestFirst, GreedyStepwise and CfsSubsetEval); AutoWeka was executed first for 20 minutes and identified as the best models those built using RCC at 7 or 8 Å with our without side chains. These same models were further analyzed under AutoWeka for 24 hours. In all these cases, random forest with hyper-parameters unlimited tree depth, no attribute selection and 100 iterations rendered the best results …”.

3) Thirdly, you provide results in the form of accuracy values per the use of RCC in being able to recoup structural and functional information about proteins. But how does the use of RCC compare to the use of, say, just course-grained PDB data in various ML-based methods to recoup structure classification and gene ontology details?

The structural classificication performance against other models was already compared in our previous work (https://www.ncbi.nlm.nih.gov/pubmed/26366526). There we showed that RCC outperform the best ML models and other geometrical approaches reported so far; we stated this in the introduction section “…such representation allowed to identify structural neighbors and classify the protein 3D structure with the best performance and reliability reported so far [11]⁠…”. In the present work we compared our results with that of our previous results only to avoid duplicating results.

In terms of comparing our results with previous work in the protein function classification, in Table 3 we present the comparison of our approach with the best models identified in CAFA2. An important aspect about functional classification is that there is no recognized unique standard to classify protein function; for instance, the Gene Ontology classification uses 3 different criteria: molecular function, biological process and cellular localization. But protein function can also be classified according to enzymatic activity, which does follow the same logic as GO. CAFA experiment represents an effort to classify and predict protein function and hence our motivation to compare our method with the results provided by this experiment. Indeed, CAFA uses a different set of proteins than ours; particularly, CAFA2 used 100,755 protein sequences of which only 15,617 sequences matched with a PDB entry, specifically with 217,409 chains on the PDB, or 71,911 different PDB entries. This is the consequence that a single UniProt entry corresponds with more than one PDB chain. Thus, basically we used 15% of the data set used in CAFA2 and reproduce the results in classification than the best CAFA models. It is important to emphasize that the best models in CAFA2 used sequence alignments and ML models that incorporated diverse proteins features (see https://genomebiology.biomedcentral.com/articles/10.1186/s13059-016-1037-6 and particularly Supplementary Figures 11A-11F and Supplementary Table 1), while our approach only used RCC to achieve these results. To clarify this, we have now added an explanation on this similarities and differences in the Results section: “…Table 3 summarizes the best model performance compared with the statistics reported at CAFA2 for the best models [20]. Fmax is the maximum harmonic representation of the precision and recall achieved by a set of models; Fmax=1 is for a perfect predictor. It is important to note that the CAFA experiment attempts to unify the prediction of protein function, but does not use protein 3D structure hence, 15% of all sequences included in CAFA2 were included in our datasets (data not shown). The best models in CAFA2 used sequence alignments and ML models that incorporated diverse proteins features, while our models only used RCC, that is, features derived exclusively from the protein structure. The purpose of this comparison is not to show better performance than CAFA2 models, but to note the level of reliability of our predictions in comparison with the known best function predictors; that is, the best predictors in CAFA2 were close to Fmax=0.5, so as our models. Hence, these results show that RCC achieved reliable classifications in both protein structure and protein function…”.

4) You mention that RCC "is able to learn protein function." Don't you mean that an ML model is able to learn?

We appreciate the comment; this was a mistake. We have changed this phrase and now we state in Introduction that “…Furthermore, we showed that ML models built from RCC were able to efficiently learn protein function…”.

5) The pseudocode in 2.1 lacks any reasonable amount of detail. Three lines only? And how do you generate a graph? What are the details?

The pseudocode detailing the graph generation was added at the end of section 2.1.1 as follows:

The pseudo-code for this calculation is detailed in Algorithm 2:

Input: Cartesian coordinates of protein atoms

Output: Contact graph

For each residue r in the protein:

     Calculate its cube in the grid Gr using the hash function.

     Get the list of residues L inside the cubes neighboring Gr.

     For each residue s in L:

                Calculate D distance between rc and sc.

                Quick inclusion if D<d and continue.

                Quick exclusion if minimum {Drd, Dsd} > d and continue.

                Perform contact check for each pair of atoms from r and s.

6) The run times for calculate RCC are impressive, but you don't mention compute resources. Use of HPC? Quad core? GPUs?

We appreciate the comments; we did not include such technical detail that is important to appreciate the relevance of our results. We added the following description in Methods and Results sections to clarify this point: “…on a 64-bit Intel-Xeon linux-based server with 24 cores and 256 GB of RAM…”

7) The use of the term "hash-like" is ambiguous. Hashing in the context of computer science refers to hash functions, which for perfect hashing are O(constant). I think there is cross-discipline terminology confusion here that should be cleared up. And what do you mean by "like." Why not call it Grid-like?

Indeed the use of hash-like term is ambiguous; we appreciate the note. To clarify this, we have now change the phrase in section 2.1.1 that refers to this operation:

  1. i) Hashing approach using a 3D grid over the space. We generated a grid over the 3D space of width d that corresponded with the contact threshold distance, thus given a residue we can identify the cube that contains it by using a hashing function; every cube in the grid has a list of all the residues that are inside of it. Given a residue r and the cube in the grid where it is found Gr, we know that any other residue inside Gr must be in contact with r. Additionally, if a residue s is in contact with r, it must be that the cubes Gr and Gs are neighbors in the grid (if they were not, then r and s must be at a distance greater than d). This way we reduced dramatically the search space (see Figure 2), which turns the complexity of the algorithm to 2*n in the linear portions of the protein and 16*n on average (given that at 5 Å a residue is in contact with 4 other residues on average).

8) For the hash-like approach, what is the cost of identifying if atoms are in the same grid? You don't provide details, and it seems like your hash-like approach is a magical fix for everything.

We did not clarify this in the first version of our work; we appreciate the note. In order to identify if two atoms are in the same cube in the grid, the hash function is used to obtain the cube number and compare them. The complexity of this calculation is constant (the hashing function just requires 3 multiplications and 2 additions which is performed two times plus the comparison)

The text in the previous question addresses this concern.

9) For your quick rules evaluation approach, how does that improve on n^2? The intro to section 2.1.1 mentions n^2, so please explain further.

The text in section 2.1.1 now elaborates more on this:

Additionally, if a residue s is in contact with r, it must be that the cubes Gr and Gs are neighbors in the grid (if they were not, then r and s must be at a distance greater than d). This way we reduced dramatically the search space (see Figure 2), which turns the complexity of the algorithm to 2*n in the linear portions of the protein and 16*n on average (given that at 5 Å a residue is in contact with 4 other residues on average).

10) Why at most are 6 residues considered? In a sphere of size 15 Angstroms, is that the maximum number of residues that can fit into that space?

The original work reporting the concept of RCC (https://www.ncbi.nlm.nih.gov/pubmed/26366526) considered a contact distance of 5 Angstroms and found clusters of 7 residues extremely rare and that adding them didn’t improve the performance. By allowing larger distances this number clearly increases, but we decided to keep it constant for two reasons: having more features increases the risk of over-fitting and by keeping it constant, the results between our previous and current results, as well as current results between themselves are comparable.

The text in section 2.1.3 was changed to elaborate on this: “…The maximum size of cliques is limited to 6 due to larger numbers being extremely rare at 5 Å and was kept as is in all the runs in order to keep the results comparable and avoid over-fitting…”.

11) Does [1,1,1,1,1,1] refer to a long loop? It would be good to get a better sense of these, much like you did for [4], [2,2], [1,2].

[1,1,1,1,1,1] corresponds to a cluster of 6 residues that has 5 protein regions between them. We referred to a similar case in Figure 1, where we describe the case [1,1,1] and in the main text in section 2.1.2 we also referred to this case as: “…For instance, a RCC with 3 residues where all residues are not adjacent in the protein sequence is referred to as [1,1,1]…”. The region separating these residues do not necessarily are loops, these regions could have secondary structures and/or loops; to clarify this, we have now added in figure 1 legend and the text in section 2.1.2: “…The regions separating these non-adjacent residues in the protein sequence could include different secondary structure elements (e.g., loops, helices)…”.

12) You mention a variant -- use of side chain atoms -- of RCC. But you never mention at the onset that your "original" RCC is coarse grained, which I presume is CA only.

The original work used the side chain atoms; the current implementation allows excluding these atoms.

The introduction section was changed to explain this better: “…The representation is based on counting the 26 different maximal clique classes that are derived from the 3D structure and protein sequence given a contact distance threshold of 5 Å including atoms of the side chains; we referred to these maximal cliques as Residue Cluster Classes or RCC (see Figure 1 and Methods). In that previous work, we tested two ML algorithms (random forest and support vector machine) that were adequate to the structural classification of the data. In the present work, we developed a computationally efficient implementation to compute RCC on large datasets of protein 3D structures. This allowed us to further explore the protein structure classification and distribute this implementation freely (see https://github.com/C3-Consensus/RCC); this implementation incorporates some variations in the contact definition (different distances and the exclusion of the side-chain atoms). Here we also explored in a systematic way dozens of ML models to identify the optimal model and corresponding hyper-parameters for such tasks and corroborated that random forest rendered the best algorithm …”.

13) Grammar-wise, often times articles ("the" and "a") are missing. 

We have revised our written work more carefully and found several problems that now are fixed.

14) I like the figures, but the text in them is too big, suggesting a not-too-careful cut and paste from a different manuscript approach.

All figures and corresponding legends are original. To address this comment and facilitate the reading of our work, several of our figures now have shorter figure legends.

Reviewer 2 Report

The manuscript describes an efficient approach to derive representation of proteins from their 3D structures, which can be further used for structural classification and function prediction. The approach seems reasonable, while, I have a few suggestions that could strengthen presentation of the article.

Major:

  1. In line 91, please provide detailed description about how the sphere radius r_d and s_d is determined. Do different residues have varied sphere radius? Which atom type is used to calculate the distance D between two residues, CA or CB only?

  1. What’s difference in calculation of contact map when the side-chain atoms are included or excluded, as shown in Figure 3? Please describe this in details.

  1. In line 128, ‘8 different distance cut-off’ is mentioned but only 7 distance values are provided (5,6,7,8,9,10,15), which is confused to figure out how the 16 RCC representations were obtained with/without side-chains included (16 by 8+8 or 14 by 7+7?). Similar statement is found in line 138, in which ‘7 cut-off values’ is mentioned.

  1. In line 140, for the function classification, how many proteins in the PDB database have annotated functions found in Gene Ontology database? How the number 4991252 is derived?

  1. In Table 3, the performance of functional classification from three different analysis may not be comparable since their benchmark datasets are completely different. Authors should test the performance on the same benchmark proteins, and further compared their methods with other function prediction approaches on the same benchmark dataset. Otherwise, it is not rigorous to state comparable efficiency of their method with sequence-based methods in CAFA2.

  1. Another concern is about the redundancy in training and testing data. Did authors perform structural similarities checking among all proteins used in this study? Highly similar structures (i.e. same family proteins) should be removed before applying 10-fold cross validation to get more robust results.

Minor:

  1. In section 2.1.1, some terms in method description of contact map calculation using 3D-grid are not clear. In line 80, is the width d set for grid over the overall 3D space, or for cube around interested residue? The terms ‘cube’ and ‘grid’ are somewhat mixed and hard to clearly understand. The better way for reader to understand the calculation is annotating the width d in the figure 2, along with highlighted cubes.

  1. Visualization of Figure 5 and Figure 6 should be improved. It will be much clearer to separate the lines by categories, for instance, comparing accuracy difference between with/without side-chain atoms for Class annotation and Architecture in two different subplots in Figure 5. Similar issues found in Figure 6.

  1. Line 96-98 may need be rephrased for better understanding why D+d is used here.

  1. In line 130, 165 and Figure 1, the number of proteins is not consistent (354079 vs 354078).

Author Response

1) In line 91, please provide detailed description about how the sphere radius r_d and s_d is determined. Do different residues have varied sphere radius? Which atom type is used to calculate the distance D between two residues, CA or CB only?

The radius is the distance between the center of mass of the residue and the atom farthest from it. Different residues may have varied radius mainly due to the size of the lateral chain and also due to the elasticity of the bonds. The distance from their centers determines the distance between two residues. Figure 3 and its legend were changed to explain this better:

Figure 3. Contact map algorithm. r and s are two residues potentially in contact, each is enveloped in a sphere with center rc and sc, and radius rd or sd, respectively. The contact distance is d and the distance between rc and sc is D. For each sphere the center is calculated as the geometric center for all the atoms in the residue including the lateral side-chain atoms (the center doesn’t necessarily overlap with any atom) and the radius is the distance between the center and the farthest atom, which may vary depending on the length and structure of the lateral chain. In the case where the lateral side-chain is ignored, the spheres may still be not identical due to the elasticity of the bonds between the atoms (i.e. the distance between the Nitrogen and Carbon alpha atoms is not exactly constant just as the internal angles of the backbone).

2) What’s difference in calculation of contact map when the side-chain atoms are included or excluded, as shown in Figure 3? Please describe this in details.

Besides changing Figure 3 and its legend, we have also added some better explanation of this matter in section 2.1.1: “…In the scenario where the lateral side-chains are ignored, the spheres are calculated with the remaining atoms of the residue. The geometric restrictions work in the same way as if the spheres were calculated with all the atoms in the residue and similarly, when a pair of residues does not meet the criteria for quick evaluation, the explicit computation of the distances between their pair of atoms is performed exclusively with the atoms in the backbone. The pseudo-code for this calculation is detailed in Algorithm 2:…”.

3) In line 128, ‘8 different distance cut-off’ is mentioned but only 7 distance values are provided (5,6,7,8,9,10,15), which is confused to figure out how the 16 RCC representations were obtained with/without side-chains included (16 by 8+8 or 14 by 7+7?). Similar statement is found in line 138, in which ‘7 cut-off values’ is mentioned.

We appreciate the note; indeed there was an inconsistency in our description. The correct number is 7 distances; the text along our work was changed to clarify this. To clarify the source of this confusion, we originally computed RCC using 9 different distances, but learned that 2 of them were not informative enough (3 and 4 Angstroms), so we decided not to report them.

4) In line 140, for the function classification, how many proteins in the PDB database have annotated functions found in Gene Ontology database? How the number 4991252 is derived?

We have changed the text in section 2.2 to explain how this number is derived: “…By cross-referencing the 869,535 By cross-referencing the 869,535 functions reported in GO (C: 176,437; F: 422,681; P: 270,416) with the 354,079 protein domains in the PDB, we obtained a database of 4,991,252 annotated proteins (C: 1,192,742; F: 2,2,85,509; P: 1,513,001) with their corresponding RCC and known protein functions. The total number of annotated proteins is higher than the PDB protein domain because several proteins have multiple annotated functions, and a single protein sequence may have multiple chains in a single PDB file …”.

5) In Table 3, the performance of functional classification from three different analysis may not be comparable since their benchmark datasets are completely different. Authors should test the performance on the same benchmark proteins, and further compared their methods with other function prediction approaches on the same benchmark dataset. Otherwise, it is not rigorous to state comparable efficiency of their method with sequence-based methods in CAFA2.

The goal of our work is not to show that our method is the best to predict correctly protein function. Instead, our goal is to show that the same protein representation is effective to classify both protein structure and function. An important aspect about functional classification is that there is no recognized unique standard to classify protein function; for instance, the Gene Ontology classification uses 3 different criteria: molecular function, biological process and cellular localization. But protein function can also be classified according to enzymatic activity, which does follow the same logic as GO. CAFA experiment represents an effort to classify and predict protein function and hence our motivation to compare our method with the results provided by this experiment. Indeed, CAFA uses a different set of proteins than ours; particularly, CAFA2 used 100,755 protein sequences of which only 15,617 sequences matched with a PDB entry, specifically with 217,409 chains on the PDB, or 71,911 different PDB entries. This is the consequence that a single UniProt entry corresponds with more than one PDB chain. Thus, basically we used 15% of the data set used in CAFA2 and reproduce the results in classification than the best CAFA models. It is important to emphasize that the best models in CAFA2 used sequence alignments and ML models that incorporated diverse proteins features (see https://genomebiology.biomedcentral.com/articles/10.1186/s13059-016-1037-6 and particularly Supplementary Figures 11A-11F and Supplementary Table 1), while our approach only used RCC to achieve these results. To clarify this, we have now added an explanation on this similarities and differences in the Results section: “…Table 3 summarizes the best model performance compared with the statistics reported at CAFA2 for the best models [20]. Fmax is the maximum harmonic representation of the precision and recall achieved by a set of models; Fmax=1 is for a perfect predictor. It is important to note that the CAFA experiment attempts to unify the prediction of protein function, but does not use protein 3D structure hence, 15% of all sequences included in CAFA2 were included in our datasets (data not shown). The best models in CAFA2 used sequence alignments and ML models that incorporated diverse proteins features, while our models only used RCC, that is, features derived exclusively from the protein structure. The purpose of this comparison is not to show better performance than CAFA2 models, but to note the level of reliability of our predictions in comparison with the known best function predictors; that is, the best predictors in CAFA2 were close to Fmax=0.5, so as our models. Hence, these results show that RCC achieved reliable classifications in both protein structure and protein function…”.

6) Another concern is about the redundancy in training and testing data. Did authors perform structural similarities checking among all proteins used in this study? Highly similar structures (i.e. same family proteins) should be removed before applying 10-fold cross validation to get more robust results.

Indeed, biased training sets in which instances of a class are over-represented (e.g., too many examples of the structural class “mainly alpha” are present in comparison with those of few secondary structure) has the potential risk to over-predict the over-represented class (e.g., most proteins would be classified as mainly alpha), yet this bias does not always render biased predictions hence it is important to test for any possible bias on the predictions. However, detecting possible bias in the prediction of multi-class problems is not as easy as for binary classes problems, because in multi-classes the false negatives are distributed over many classes. An alternative approach is the use of the ZeroR algorithm, which simply predicts all classes as the one with more instances (the biased class). Since functional classification was based on a structural representation of proteins (that is, the RCC), performing an analysis on the functional annotation may reveal any bias in both structure and function classifications. We performed this analysis for the best model using RCC built from a distance of 7 A in cellular localization; the number of classes in this case is 84, thus we expect that each class would be represented in 1.2% if evenly represented. The ZeroR method rendered 12% of correctly classified instances, while the best model predicted 39% of correctly classified instances, indicating that the bias did not negatively affect the model efficiency to classify cellular localization. We performed this analysis for all distances and all functional classifications in GO (cellular localization, molecular function and biological process) and present these results in Figure 7. Accompanying text is also included to explain the purpose and results obtained in this analysis. In summary, our results were not biased by the biased composition in the training sets.

7) In section 2.1.1, some terms in method description of contact map calculation using 3D-grid are not clear. In line 80, is the width d set for grid over the overall 3D space, or for cube around interested residue? The terms ‘cube’ and ‘grid’ are somewhat mixed and hard to clearly understand. The better way for reader to understand the calculation is annotating the width d in the figure 2, along with highlighted cubes.

The width d is set for the grid over all the 3D space. The text has been adjusted to avoid confusion between the grid and the cubes in it.

  1. i) Hashing approach using a 3D grid over the space. We generated a grid over the 3D space of width d that corresponded with the contact threshold distance, thus given a residue we can identify the cube that contains it by using a hashing function; every cube in the grid has a list of all the residues that are inside of it. Given a residue r and the cube in the grid where it is found Gr, we know that any other residue inside Gr must be in contact with r. Additionally, if a residue s is in contact with r, it must be that the cubes Gr and Gs are neighbors in the grid (if they were not, then r and s must be at a distance greater than d). This way we reduced dramatically the search space (see Figure 2), which turns the complexity of the algorithm to 2*n in the linear portions of the protein and 16*n on average (given that at 5 Å a residue is in contact with 4 other residues on average).

8) Visualization of Figure 5 and Figure 6 should be improved. It will be much clearer to separate the lines by categories, for instance, comparing accuracy difference between with/without side-chain atoms for Class annotation and Architecture in two different subplots in Figure 5. Similar issues found in Figure 6.

We have changed Figures 5 and 6 to address this concern and their legends. We appreciate the suggestion.

9) Line 96-98 may need be rephrased for better understanding why D+d is used here.

was a mistake, the correct distance is d. We have changed the description of the test referring to this distance in section 2.1.1 as follows: “…Finally, they must be in contact if a sphere of radius d and center rc overlaps half the volume of the sphere enveloping s or vice versa (though this last rule was turned off in the final implementation due to the overhead of the calculation being too similar to the time saved by it). Any pair of residues that do not fulfill one of these rules needs the explicit computation of the distance between its pair of atoms to be performed (with an early stop if a contact is found)…”.

10) In line 130, 165 and Figure 1, the number of proteins is not consistent (354079 vs 354078).

The correct number is 354,079, it has been corrected in the text of our current version.

Reviewer 3 Report

This paper proposed a new concept, RCC (Residue Cluster Classes), which are 26 different maximal clique classes from the 3D structure and protein sequence. RCC calculation includes 3 steps: contact map calculation, maximal cliques calculation and RCC calculation. And RCC can be used to classify protein structures and functions. 1. The idea of RCC is very creative and different from traditional protein classification. 2. It makes sense that the RCC approach can be used for both structural and functional classification. 3. I checked the data on github. In my understanding, some proteins have deleted some residues. What is the reason for the deleted residues? 4. The meaning of data and programs is not clear. My understanding of the data is that each protein uses a 26-dimenional vector represent the number of each RCC. Maybe you should explain what the data means and how the programs work. 5. In "data" parts, I have one specific questions: Why are some residues in some proteins deleted? 6. The format of formula 1-3 is not uniform or not alignment. 7. The format of references should be revised. (please read "Instructions for Authors" in journal website)

Author Response

1) I checked the data on github. In my understanding, some proteins have deleted some residues. What is the reason for the deleted residues?

As a sample from the output

103lA00         [8, 10, 7, 0, 4, 2, 14, 5, 0, 0, 0, 1, 1, 11, 80, 0, 0, 0, 0, 0, 0, 0, 0, 0, 0, 4]    157      5          35|36|37|38|39

The first column includes the protein ID (1031) and its chain (A) followed by the corresponding domain (00), the second column is the RCC, then the number of residues included in the domain (157 in this case) followed by the missing residues (5) and the corresponding positions of the missing residues. In this example, the 5 missing residues are after residue 34, and the protein continues in residue 40. Frequently missing residues in PDB entries are the consequence that in this region it was not possible to establish the electronic density of atoms. To clarify this, we have now added some explanation in the GitHub web site.

2) The meaning of data and programs is not clear. My understanding of the data is that each protein uses a 26-dimenional vector represent the number of each RCC. Maybe you should explain what the data means and how the programs work.

We assumed this concerns raises in the Discussion section: “…In terms of computing time, assuming these calculations would be done on a single central processing unit or CPU, the global time required to compute RCC is divided in the time required to load data (elapsed time) and the time the program takes to be executed (CPU time)…”. So, we changed this text to clarify this better: “…In terms of computing time, assuming these calculations would be done on a single central processing unit or CPU, the global time required to compute a given RCC is divided in the time required to load the protein 3D structure or PDB file (elapsed time) and the time to obtain the RCC (CPU time) that includes constructing the contact map of residues and identifying the maximal cliques. Our results indicate that most of the time is taken by reading the PDB file …”.

3) In "data" parts, I have one specific questions: Why are some residues in some proteins deleted?

These residues were not deleted, but are missing from the original PDB entry, most likely as a consequence of lack of sufficient experimental evidence to localize the atoms of those residues.

To clarify this situation, the README.md has been updated in the github repository as follows:

# RCC

The RCC is a 26-dimensional vector, each value corresponding to a feature from the topology of the protein: the number of residues of a given cluster class.

Options to run code:

            -pdb                                <file to be processed>

            -chain                            <chain in the pdb file to be processed>

            -pathTomita               <path to tomita executable file>

            -lateralChain   <yes/no (no)>

            -minDist                        <minimum contact distance (0)>

            -maxDist                       <maximum contact distance (7)>

            -fileList                       <csv file with filename, chain>

            -pathPdbFolder          <path to pdb database folder>

The program will output the RCC on the terminal with the format:

<Protein> [<RCC separated by commas>] <Number of residues considered> <Number of residues missing> <Ids of the missing residues>

e.g. 103lA00   [8, 10, 7, 0, 4, 2, 14, 5, 0, 0, 0, 1, 1, 11, 80, 0, 0, 0, 0, 0, 0, 0, 0, 0, 0, 4]    157      5          35|36|37|38|39

For the protein 103l its RCC is [8, 10, 7, 0, 4, 2, 14, 5, 0, 0, 0, 1, 1, 11, 80, 0, 0, 0, 0, 0, 0, 0, 0, 0, 0, 4], it has 157 residues and 5 residues missing in the PDB file (35, 36, 37, 38, 39).

4) The format of formula 1-3 is not uniform or not alignment.

The formulas have been centered and due to another correction in the text, the formula (3) was removed.

5) The format of references should be revised. (please read "Instructions for Authors" in journal website)

We have revised the format of the references and made the required adjustments.

Round 2

Reviewer 1 Report

Minor edits only. Most of the below are grammatical, and I hope are addressed by the editorial staff just prior to publication.

In the abstract, change "such relationship" to "such a relationship".

In the abstract, change "three-dimensional structure and show that such representation" to "three-dimensional structures and show that such representations".

Line 46, "proteins 3D structure" is grammatically incorrect. Protein should be singular, or structure should be plural.

Line 102, fix the typo "5 A a"

Line 118, please make sure (2) is on the right side of the page and not the left

Figure 3, and lines 110-125, in the figure, the "c" and "d" designators of r and subscripts, but in lines 110-125, the "c" and "d" are just smaller font. Please be consistent.

Figure 3 and lines 128-130, the "d" part of rd and sd is confusing, because "d" usually refers to diameter.

Line 129, confusing that "the contact distance is d" because the "d" for r is different than the "d" for s. Shouldn't you say that the contact distance is rd and sd?

Line 137, what is a "lateral" side chain?

Linr 220, change "allowed us to compute the whole PDB database" to "allowed us to analyze all of the domains in the PDB" or "allowed us to compute RCC values for all protein structures in the PDB."

Lines 264-265, what type of cross validation? 10-fold? 5-fold? Something else?

Line 340, change to "the CAFA contest."

Line 360, change "codes" to "code."

Reviewer 2 Report

Authors have addressed all my comments.

Reviewer 3 Report

My questions have been answered.